# Ginsenoside Re Ameliorates UVB-Induced Skin Photodamage by Modulating the Glutathione Metabolism Pathway: Insights from Integrated Transcriptomic and Metabolomic Analyses

**DOI:** 10.3390/ijms27020708

**Published:** 2026-01-10

**Authors:** Jiaqi Wang, Duoduo Xu, Yangbin Lai, Yuan Zhao, Qiao Jin, Yuxin Yin, Jinqi Wang, Yang Wang, Shuying Liu, Enpeng Wang

**Affiliations:** Ginseng Academy, Changchun University of Chinese Medicine, Changchun 130117, China; 23203070307@stu.ccucm.edu.cn (J.W.); czxuduoduo@163.com (D.X.);

**Keywords:** ginsenoside Re, skin photodamage, ultraviolet radiation, glutathione metabolism, multi-omics analysis, natural photoprotectants

## Abstract

With the growing prominence of skin photodamage caused by ultraviolet (UV) radiation, the development of efficient and safe natural photoprotectants has become a major research focus. Ginsenoside Re (G-Re), a primary active component of ginseng (*Panax ginseng* C. A. Mey.), has attracted much attention due to its significant antioxidant and anti-inflammatory activities; however, its systemic role and mechanism in protecting against photodamage remain unclear. In this study, a UVB-induced rat photodamage model was established to evaluate the protective effect of ginsenoside Re through histopathological staining, biochemical assay, and immunohistochemical analysis. Furthermore, an integrated transcriptomic and metabolomic approach was applied to elucidate the molecular mechanism of G-Re protection and to establish the association between the photodamage phenotype, metabolic pathways, and gene functions. Following their identification via integrated multi-omics analysis, the key targets were subjected to verification via Western blotting. The results showed that G-Re could effectively alleviate UVB-induced pathological injury and reduce the level of oxidative stress and inflammatory factors, which could reverse regulate the abnormal expression of 265 differential genes and 30 metabolites. The glutathione metabolism pathway was proven as a key pathway mediating the protective effects of ginsenoside Re against skin photodamage via integrated analysis, WB verification, and molecular docking. The current study indicated that G-Re could be a promising natural sunscreen additive in cosmetical products.

## 1. Introduction

In modern society, accelerated industrialization, environmental changes, and evolving lifestyles have exacerbated ozone layer depletion. Approximately 30–40% of the global population is at risk of photodamage, especially long-term outdoor workers and residents in high-ultraviolet regions [1,2]. Cutaneous photodamage is a common skin condition caused by UV radiation. Prolonged exposure to UV radiation triggers a series of complex biological reactions in the skin, resulting in erythema, wrinkles, pigmentation, and potentially even serious consequences such as skin cancer [3]. Based on wavelength differences, UV radiation is categorized into short-wave ultraviolet (UVC), medium-wave ultraviolet (UVB), and long-wave ultraviolet (UVA) [4]. Among these, UVC has the shortest wavelength and highest energy but is typically absorbed by the ozone layer, preventing it from reaching the Earth’s surface. UVB represents the most biologically active component of sunlight, with phototoxicity approximately 1000 times that of UVA (320–400 nm); it induces classic erythema and sunburn cells, serving as a key driver of acute skin injury [5]. While long-wave UVA does not cause acute skin inflammation, its effects on the skin are gradual and cumulative over time, contributing to skin aging and damage [6]. Thus, preventing skin harm from UV radiation primarily involves blocking the UVB spectrum. The underlying pathogenic mechanism mainly involves excessive reactive oxygen species (ROS) generation induced by UVB skin exposure, which causes DNA damage, collagen degradation, and inflammatory factor release—ultimately accelerating photoaging and even triggering skin tumors [7,8]. Currently, most photoprotective products rely on physical or chemical sunscreens, which suffer from issues like poor skin permeability or side effects. Consequently, the discovery of efficient and safe photoprotectants from natural products has emerged as a key research direction [9,10].

Ginseng, a precious traditional Chinese medicinal herb with thousands of years of application in traditional Chinese medicine, contains ginsenosides as its core active components, which exhibit extensive biological activities. Ginsenoside Re, a primary ginsenoside, may offer advantages such as high skin permeability and low irritation owing to its relatively small molecular size and natural origin. Recent studies have demonstrated that ginsenoside Re possesses multiple biological activities, including antioxidant, anti-aging, and immunomodulatory effects [11,12]. These properties align closely with the key mechanisms underlying UV-induced skin damage, suggesting that ginsenoside Re holds significant potential for preventing skin photodamage [8]. However, systematic investigations into the pharmacological effects, molecular mechanisms, and global regulatory networks of ginsenoside Re in protecting against photodamage remain limited, warranting further exploration using modern research methodologies. Over recent decades, the rapid advancement of emerging omics technologies—particularly transcriptomics and metabolomics—has provided powerful tools for systematically studying complex biological processes under drug intervention. Metabolomics enables comprehensive profiling of global changes in small-molecule metabolites within organisms, directly reflecting metabolic phenotypes and functional endpoints under physiological or pathological conditions [13,14]. In contrast, transcriptomics focuses on analyzing qualitative and quantitative dynamics of mRNA molecules at the gene expression level, elucidating key mechanisms of upstream regulatory events and signaling pathways [15]. The integration of metabolomics and transcriptomics enables multi-level information integration from “genetic instructions” to “metabolic responses”, providing a solid foundation for systematically elucidating the mechanisms of drug action [16].

Based on the above background and technical strategies, this study established a UVB-induced rat model of skin photodamage to systematically evaluate the protective effects of ginsenoside Re against UV-induced skin photodamage. By integrating multi-omics approaches to deeply investigate its molecular mechanisms, this research aims to provide systematic and in-depth experimental evidence for the anti-photodamage mechanisms of ginsenoside Re. This not only helps expand the application of ginsenosides in the field of skin photoprotection but also provides a theoretical basis for developing new natural anti-photoaging drugs.

## 2. Results

### 2.1. Protective Effects of G-Re on UVB-Irradiated HaCaT Cells

As shown in Figure 1A, treatment with G-Re at concentrations ranging from 5 to 90 µg/mL for 24 h did not induce significant cytotoxicity in normal HaCaT cells, as determined by MTT assay. No statistically significant differences in cell viability were observed between any G-Re-treated group and the normal control group (*p* > 0.05). To evaluate the protective effect of G-Re against UVB-induced damage, an in vitro photodamage model was established. As illustrated in Figure 1B, HaCaT cell viability decreased in a dose-dependent manner with increasing UVB irradiation doses (60–480 mJ/cm^2^). However, co-treatment with G-Re significantly enhanced cell survival under all tested UVB doses. Specifically, at a UVB dose of 60 mJ/cm^2^, cell viability began to improve significantly with 5 µg/mL G-Re compared to the model group (*p* < 0.05), and a more pronounced protective effect was observed at 45 µg/mL G-Re *(p* < 0.01). Even at higher UVB irradiation doses (120, 240, and 480 mJ/cm^2^), high concentrations of G-Re (90 µg/mL) significantly reversed the UVB-induced reduction in cell viability (*p* < 0.05 or *p* < 0.01 compared to the corresponding model groups). To investigate the regulatory effect of ginsenoside Re on UVB-induced oxidative stress in skin cells, we determined the intracellular content of malondialdehyde (MDA), a hallmark of lipid peroxidation, and the activity of superoxide dismutase (SOD), a key antioxidant enzyme. As shown in Figure 1C, compared with the blank control group, UVB irradiation significantly suppressed SOD activity (*p* < 0.01) and increased MDA content (*p* < 0.01) in HaCaT cells. In contrast, G-Re intervention significantly elevated SOD activity and reduced MDA levels (*p* < 0.01) compared to the UVB model group, with a trend toward greater improvement at higher doses. These findings suggest that G-Re may alleviate UVB-induced oxidative damage by enhancing cellular antioxidant capacity and inhibiting lipid peroxidation.

### 2.2. Histopathological Analysis of Skin Tissues by H&E and Masson Staining

As shown in the H&E staining of rat skin (Figure 1D), the control group exhibited an intact and relatively thin epidermal structure with clearly stratified keratinocytes. In contrast, UVB-irradiated rats showed marked epidermal hyperplasia, disordered cell arrangement with atypia, and substantial inflammatory cell infiltration in the dermis. Treatment with ginsenoside Re attenuated these alterations in a manner corresponding to the administered dose, resulting in reduced epidermal thickness and diminished inflammatory infiltration. Masson’s trichrome staining of rat skin (Figure 1E) revealed that the dermis of control animals contained abundant, densely packed, and well-aligned blue collagen bundles. UVB exposure significantly reduced collagen content, accompanied by fragmentation, disorganization, and thinning of the residual collagen fibers. Areas of elastic fiber degeneration, appearing as light pink masses, were also observed, occupying spaces originally filled by collagen. Administration of ginsenoside Re ameliorated these structural abnormalities, with greater improvement observed at higher doses. In summary, micromorphological analysis via H&E and Masson’s trichrome staining confirmed that ginsenoside Re alleviated UVB-induced epidermal hyperplasia, cellular atypia, and dermal inflammatory infiltration in rat skin, showing enhanced efficacy with increasing dose. It also significantly preserved collagen fiber structure and suppressed elastic fiber degeneration, demonstrating a clear protective and restorative effect against UVB-induced skin photodamage.

### 2.3. Immunohistochemical Analysis of COX-2 and MMP-9 in Skin Tissue

Immunohistochemical analysis (Figure 1F) was quantified by measuring the percentage of positive area (Area%) using Image J software (version 1.54), with a higher Area% indicating increased protein expression level. UVB irradiation significantly upregulated the protein expression of both COX-2 and MMP-9 in rat skin tissues compared to the control group (*p* < 0.05), confirming the successful establishment of the photodamage model. Ginsenoside Re intervention exhibited a dose-dependent protective effect. Notably, the high-dose ginsenoside Re group significantly suppressed the abnormal overexpression of both COX-2 and MMP-9 (*p* < 0.01 vs. model group) and restored their expression levels to those statistically comparable to the control group (*p* > 0.05). These results demonstrate that high-dose ginsenoside Re completely reverses UVB-induced aberrant expression of key inflammatory (COX-2) and matrix-degrading (MMP-9) proteins.

### 2.4. Biochemical Evaluation of Skin Tissues

To investigate the ameliorative effects of ginsenoside Re on UVB-induced inflammation and oxidative stress in skin photodamage, we measured the levels of CAT, SOD, GSH-Px, MDA, T-AOC, IL-6, IL-8, and TNF-α in rat skin tissues. In UVB-induced rats, compared with the control group, the levels of IL-6, IL-8, TNF-α, and MDA were significantly increased, while the activities of CAT, SOD, GSH-Px, and T-AOC were markedly decreased (Figure 1G). These changes were statistically significant (*p* < 0.01). Treatment with ginsenoside Re ameliorated these inflammatory and oxidative stress markers, with more pronounced effects observed at higher doses.

### 2.5. Organ-to-Body Weight Ratio of Rats

To evaluate the systemic effects of UVB radiation and ginsenoside Re intervention on major organs in rats, the organ-to-body weight ratios (organ weight/body weight) were measured on day 14 of the experiment, and the results are presented in Figure 1H. Statistically significant differences among the groups were primarily observed in the lung, spleen, and thymus. As shown in Figure 1H, the lung index was significantly increased in the UVB model group compared with the blank control group (*p* < 0.01), suggesting that prolonged UVB exposure may induce pulmonary inflammation and edema, thereby increasing lung weight. In contrast, G-Re treatment significantly reduced the lung index compared to the UVB model group (*p* < 0.01), indicating that G-Re markedly ameliorates the UVB-induced elevation in lung index. The spleen index was significantly decreased in the UVB model group relative to the blank control group (*p* < 0.01), while G-Re administration resulted in a significant increase in the spleen index compared with the UVB model group (*p* < 0.05). Regarding the thymus index, UVB irradiation led to a significant reduction in normal rats (*p* < 0.05), whereas high-dose G-Re treatment restored the thymus index to a level comparable to that of the control group (*p* < 0.05). These findings collectively demonstrate that the protective effect of G-Re against UVB-induced photodamage is not confined to local skin tissues but also extends to the mitigation of systemic damage, likely through modulation of systemic immune and inflammatory status.

### 2.6. Untargeted Metabolomic Data Analysis

A non-targeted metabolomics study was conducted using the control group, high-dose treatment group (40 mg/kg), and UVB model group. The stability of the liquid chromatography-tandem mass spectrometry (LC-MS/MS) analytical platform was evaluated by examining the score plots derived from principal component analysis (PCA) and orthogonal partial least squares-discriminant analysis (OPLS-DA) models. The PCA score plots demonstrated high stability of quality control (QC) samples in both positive and negative ion modes, indicating that the LC-MS/MS system remained stable throughout the entire analytical sequence (Figure 2A,B). The OPLS-DA plots revealed clear separations between the control group and the model group, as well as between the model group and the high-dose Re treatment group, in both positive and negative ion modes, confirming the successful establishment of the model (Figure 2C–F). Permutation tests (*n* = 200) were performed for both ion modes, and the results (Figure 2G–J) indicated that the OPLS-DA models were reliable without overfitting, and thus suitable for screening potential differential metabolites. Differential metabolites were preliminarily identified based on variable importance in projection (VIP) > 1 and *p* < 0.05 from the OPLS-DA model. After treatment with ginsenoside Re, a subset of metabolites in the treatment group showed significant reversal trends when compared to the model group, suggesting that these reversed metabolites may represent key differential compounds. By matching mass spectral features against the HMDB public database, a total of 30 potential differential metabolites exhibiting reversal after high-dose Re treatment group were identified. These included dehydroascorbate, 1-methylnicotinamide, 13-hydroxyhexadecanoic acid, phytosphingosine, and choline, among others. Detailed regulatory information is provided in Table 1.

Based on the regulatory profiles of the potential differential metabolites, a heatmap was generated to visualize the relative abundance of these metabolites in ginsenoside Re-treated UVB-exposed rats (Figure 2K). Cluster analysis revealed clear separation among the experimental groups, with the Re treatment group clustering more closely to the control group, indicating a reversal effect of ginsenoside Re on UVB-induced metabolic alterations. Further analysis showed that among the 30 differential metabolites, 15 were upregulated in the model group and significantly downregulated after ginsenoside Re treatment, while the other 15 were downregulated in the model group and markedly upregulated after treatment. These results suggest that ginsenoside Re effectively restores UVB-induced metabolic disturbances and ameliorates photo-damage-associated metabolic imbalance. To investigate the potential metabolic pathways involved, the 30 reversed metabolites were subjected to KEGG pathway enrichment analysis using MetaboAnalyst 5.0. As shown in Figure 2L, KEGG pathway enrichment analysis based on the 30 rescued differential metabolites identified 25 relevant pathways. Among them, 8 pathways demonstrated statistical significance (*p* < 0.05). These significantly enriched pathways were primarily involved in amino acid metabolism (including histidine metabolism, and glycine, serine, and threonine metabolism), energy metabolism (encompassing the citrate cycle (TCA cycle) and pantothenate and CoA biosynthesis), one-carbon pool by folate metabolism, as well as glutathione metabolism. These results suggest that ginsenoside Re may exert its protective effect against UVB-induced skin photodamage by restoring the homeostasis of the aforementioned metabolic pathways.

### 2.7. Transcriptomic Data Analysis

To investigate the molecular mechanisms of UVB-induced photodamage in rats, total RNA was extracted from the skin of control, UVB-induced model, and high-dose ginsenoside Re (40 mg/kg) treated groups. Compared to the UVB model group, the control group showed significant expression changes in 1750 genes, including 548 upregulated and 1202 downregulated genes. The high-dose ginsenoside Re group exhibited significant changes in 938 genes compared to the UVB model group, comprising 538 upregulated and 400 downregulated genes (Figure 3A,B). Venn analysis identified 369 differentially expressed genes (DEGs) common to both the model vs. control and high-dose Re vs. model comparisons. Among these, 265 genes showed expression reversal after ginsenoside Re treatment. As shown in Figure 3C, to visually evaluate the rescuing effect of ginsenoside Re on the transcriptome, cluster analysis was performed on these 265 reversed DEGs. As shown in Figure 3D, clear separation was observed between the model group and the control group, confirming that UVB irradiation caused significant gene dysregulation. The gene expression profile of the Re-treated group closely resembled that of the control group, suggesting that these reversed genes may contribute to reversing UVB-induced skin photodamage. To further explore the functions of these reversed DEGs, Gene Ontology (GO) enrichment analysis was performed. As it was shown in Figure 3E, the biological process (BP) analysis revealed that the differential genes were mainly enriched in processes related to skin structure maintenance, stress response, inflammation regulation, and cellular function modulation. The cellular component (CC) terms were primarily associated with extracellular structures, cellular projections, and organelle lumina. Molecular functions (MF) were related to extracellular matrix structural constituents, enzyme activity, molecular binding, and signaling molecule activity. To further identify the core signaling pathways involved, KEGG pathway enrichment analysis was performed on the reversed differentially expressed genes. Figure 3F displays the significantly enriched pathways (*p* < 0.05). The analysis found that the differential genes were mainly enriched in the following categories: lipid metabolism-related pathways (alpha-Linolenic acid metabolism, Linoleic acid metabolism, Ether lipid metabolism, Glycerophospholipid metabolism), the inflammatory regulation pathway Arachidonic acid metabolism, cancer pathways (Hedgehog signaling pathway, Proteoglycans in cancer, Pathways in cancer), and key signaling pathways (Calcium signaling pathway, AGE-RAGE signaling pathway in diabetic complications, Ras signaling pathway). These pathways are closely related to the core mechanisms of UVB-induced skin photodamage, including metabolic disorders, inflammatory responses, abnormal cell proliferation, and signaling imbalances.

### 2.8. Integrated Analysis of Transcriptomics and Metabolomics

To systematically elucidate the therapeutic mechanism of ginsenoside Re, integrated KEGG pathway enrichment analysis was performed on the screened differentially expressed genes (DEGs) and differential metabolites (DMs) using the Metabo Analyst 5.0 platform. The results (Figure 4A) revealed significant enrichment of multiple pathways, including α-linolenic acid metabolism, linoleic acid metabolism, arachidonic acid metabolism, glycerophospholipid metabolism, glutathione metabolism, and pantothenate-CoA metabolism, indicating that ginsenoside Re exerts multifaceted regulatory effects on UVB-induced cutaneous metabolic disturbances. For selecting key pathways for further validation, we prioritized those with strong support from both multi-omics evidence and pathway significance. Although certain pathways, such as α-linolenic and linoleic acid metabolism, showed high enrichment significance, their alterations were observed only at a single omics level (either genes or metabolites alone). The pantothenate-CoA metabolism pathway was co-enriched with one DEG (*Bcat1*) and one DM (pantothenic acid), yet its statistical significance remained lower than that of the glutathione metabolism pathway. In contrast, the glutathione metabolism pathway was found to contain both DEGs (*Chac1* and *Gsta1*) and DMs (pyroglutamic acid and L-cysteine), exhibited high significance in the joint enrichment analysis, and demonstrated a complete “gene–metabolite” regulatory network (Figure 4B). Therefore, it was selected as the core pathway for subsequent functional validation. Further analysis of this pathway indicated that, compared to the model group, ginsenoside Re treatment significantly downregulated *Chac1* expression, upregulated *Gsta1* expression, and concurrently reduced the levels of pyroglutamic acid and L-cysteine (Figure 4C–F). These findings suggest that ginsenoside Re may alleviate UVB-induced skin photodamage by modulating the glutathione metabolism pathway, with *Chac1* and *Gsta1* potentially serving as key genes in this regulatory process.

### 2.9. Western Blot and Molecular Docking Analysis

To further validate the regulatory effect of ginsenoside Re on the glutathione metabolic pathway, an in-depth analysis was conducted using Western Blot to examine the expression of *Chac1* and *Gsta1* genes, as well as the core proteins GCLM and GPX1 in this pathway (Figure 4G). As shown in Figure 4H–K, compared with the Control group, the protein expression levels of GSTA1, GCLM, and GPX1 were significantly downregulated (*p* < 0.05) in the UVB model group, while CHAC1 expression was markedly upregulated. After treatment with ginsenoside Re, the expression levels of all four proteins were restored to near-normal levels (*p* < 0.05). These results, spanning multi-omics correlation to protein validation, demonstrate that ginsenoside Re significantly reverses UVB-induced disruption of key metabolic pathways. In particular, by upregulating the expression of critical proteins such as GCLM and GPX1 in the glutathione metabolic pathway, ginsenoside Re plays a central role in enhancing cutaneous antioxidant defense. Molecular docking results further revealed that ginsenoside Re exhibits strong binding affinity with CHAC1, GSTA1, GCLM, and GPX1, with binding energies of −11.2 kcal/mol, −11.0 kcal/mol, −6.9 kcal/mol, and −10.2 kcal/mol, respectively (Figure 4L–O). These findings provide structural evidence supporting the hypothesis that ginsenoside Re may directly interact with key proteins in the glutathione metabolic pathway, thereby modulating their function and influencing the skin’s antioxidant defense system.

## 3. Discussion

The present study, through integrated metabolomic and transcriptomic analysis, identified the glutathione pathway as a key regulatory target through which ginsenoside Re (G-Re) exerts its protective effects. This finding is consistent with the established role of glutathione metabolism as a central defense mechanism against UVB-induced oxidative stress in the skin [17,18]. Under normal physiological conditions, the intracellular ratio of reduced glutathione (GSH) to oxidized glutathione (GSSG) is maintained in a balanced state. This redox equilibrium regulates antioxidant defense system activity, preserves mitochondrial functional integrity, and supports efficient DNA repair, thereby providing essential molecular protection for skin cells against environmental oxidative stress [19,20]. However, UVB irradiation triggers an immediate burst of intracellular reactive oxygen species (ROS), leading to excessive consumption of GSH through its conversion to GSSG. This results in a sharp decline in the GSH/GSSG ratio, ultimately causing the collapse of the cutaneous antioxidant defense system and widespread oxidative damage [21]. To further investigate the regulatory role of G-Re on this pathway, the protein expression levels of CHAC1, GSTA1, GCLM, and GPX1 were examined by Western Blot analysis.

In the glutathione metabolic pathway, CHAC1 serves as a key enzyme responsible for glutathione degradation. This enzyme catalyzes the decomposition of glutathione into glutamate and cysteinyl glycine, leading to rapid depletion of intracellular GSH levels. Furthermore, it amplifies cellular stress signals and initiates apoptotic programs, consistent with previous studies establishing CHAC1′s role in mediating oxidative stress damage [22,23]. Ginsenoside Re was found to downregulate CHAC1 expression, thereby directly suppressing glutathione degradation to reduce its loss while indirectly blocking the UVB-induced stress and apoptosis cascade, ultimately alleviating skin cell apoptosis. This regulatory effect was corroborated at the metabolic level: metabolomics data revealed that ginsenoside Re intervention significantly reduced pyroglutamic acid levels [24], a downstream metabolite of CHAC1-mediated degradation. This metabolic evidence confirms the effective inhibition of GSH degradation. Given that GSH pool stability is essential for apoptosis suppression [25], these metabolic changes provide indirect support for ginsenoside Re’s role in mitigating UVB-induced cell apoptosis. Simultaneously, GCLM functions as a regulatory subunit of the rate-limiting enzyme in glutathione synthesis, with its expression level directly determining glutathione production efficiency [26]. Ginsenoside Re intervention significantly enhanced GCLM expression, thereby restoring GSH synthesis capacity. This regulatory effect was further validated metabolically: following ginsenoside Re administration, L-cysteine levels—a key precursor in glutathione biosynthesis—were markedly decreased [27,28], indicating its accelerated conversion into glutathione driven by enhanced GCLM activity. The combined evidence from both enzyme expression changes and metabolic substrate consumption demonstrates that ginsenoside Re promotes L-cysteine conversion into glutathione through GCLM upregulation, thereby enhancing de novo glutathione synthesis and fundamentally ensuring the restoration of intracellular antioxidant reserves. This “synthesis promotion” effect synergizes with the aforementioned CHAC1-mediated “degradation inhibition,” collectively elevating GSH levels and restoring the GSH/GSSG redox homeostasis, thereby providing an essential foundation for the efficient operation of downstream antioxidant enzyme systems. Notably, beyond ensuring adequate glutathione reserves, ginsenoside Re achieves functional synergy through additional mechanisms. GPX1, a crucial enzyme for ROS clearance in the pathway, utilizes glutathione as a substrate to efficiently reduce hydroperoxides, serving as a core executor in blocking oxidative stress damage [29]. Meanwhile, GSTA1 governs glutathione functional output; as a key Phase II detoxification enzyme, its primary function involves catalyzing the conjugation of glutathione with electrophilic oxidation products generated under UVB exposure, facilitating the formation of non-toxic conjugates for cellular export, thereby enhancing glutathione-mediated clearance of oxidative damage products [30,31]. Ginsenoside Re upregulation of GSTA1 expression not only augmented detoxification capacity but also prevented further glutathione depletion by promoting the binding and elimination of toxic metabolites, establishing a self-reinforcing protective cycle. These coordinated changes at the protein level collectively indicate that ginsenoside Re systematically modulates the glutathione metabolic pathway. This multi-target regulatory pattern was ultimately reflected in enhanced cellular antioxidant capacity: significant elevations in key antioxidant enzyme activities (including CAT, GSH-Px, and SOD) in skin tissues, coupled with a marked reduction in the lipid peroxidation end product MDA, jointly confirmed the effective restoration of the organism’s antioxidant defense system [32,33,34]. Therefore, ginsenoside Re systematically enhances cellular antioxidant capacity by coordinately regulating multiple facets of glutathione metabolism, including synthesis, degradation, detoxification, and clearance. Furthermore, molecular docking results demonstrated stable binding interactions between ginsenoside Re and these target proteins, suggesting that its regulatory influence on the glutathione pathway may originate from direct molecular interactions with key proteins.

Beyond the glutathione pathway, integrated metabolomic and transcriptomic analyses identified several additional key pathways potentially involved in the protective effects of G-Re against UVB-induced skin photodamage. These include lipid metabolism pathways—such as α-linolenic acid metabolism, linoleic acid metabolism, and glycerophospholipid metabolism—as well as arachidonic acid metabolism. It is noteworthy that these pathways are functionally interconnected with glutathione metabolism. UVB-induced reactive oxygen species (ROS) represent a common damaging factor that disrupts both glutathione homeostasis and the functionality of these pathways [35,36]. By restoring glutathione metabolism and enhancing ROS scavenging capacity, G-Re may reduce ROS-mediated damage to these pathways, suggesting an indirect protective and regulatory role of G-Re in maintaining their functional homeostasis.

Within the lipid metabolic network modulated by G-Re, the α-linolenic acid metabolism, linoleic acid metabolism, and glycerophospholipid metabolism pathways collectively constitute a core lipid system essential for maintaining skin structural integrity and functional homeostasis [37]. α-Linolenic acid and linoleic acid, as essential polyunsaturated fatty acids, serve as necessary precursors for the synthesis of key barrier components in the stratum corneum, such as ceramides and free fatty acids, which are critical for preserving the skin’s physical barrier function [38]. Glycerophospholipids, on the other hand, not only form the fundamental structural components of all cellular biomembranes but also contribute to the lamellar structure of the skin lipid barrier, thereby collectively maintaining skin permeability barrier function and membrane stability [39]. In this study, UVB radiation was shown to induce oxidative stress and generate ROS, which not only triggered the peroxidative degradation of essential fatty acids—key raw materials for barrier formation—leading to their depletion, but also directly damaged pre-existing glycerophospholipid membrane structures [40,41]. These alterations ultimately manifested as typical pathological features of photodamage, including epidermal hyperplasia and collagen disorganization, as observed in H&E and Masson staining. The significant enrichment of G-Re-responsive genes in these three lipid metabolism pathways, coupled with the observed restoration of UVB-induced skin damage, suggests that Ginsenoside Re may effectively repair UVB-impaired skin barrier function through coordinated regulation of this lipid metabolic network. Furthermore, the arachidonic acid metabolism pathway was also significantly enriched. This pathway plays a central role in the synthesis of classic pro-inflammatory mediators such as prostaglandins (PGs) and leukotrienes (LTs), and is critically involved in the inflammatory response induced by UVB [42]. Mechanistically, UVB radiation triggers substantial ROS production in skin tissue, which specifically activates phospholipase A2 (PLA2) [43]. Activated PLA2 hydrolyzes membrane phospholipids, leading to the release of arachidonic acid (AA) into the cytoplasm [44]. Free AA is then efficiently converted into key pro-inflammatory mediators, including prostaglandin E2 (PGE2), via the cyclooxygenase (COX) pathway [45]. PGE2 acts directly on keratinocytes and infiltrating immune cells, inducing the expression and secretion of inflammatory cytokines such as IL-6, IL-8, and TNF-α, thereby initiating and amplifying the inflammatory cascade and exerting systemic effects on distant organs and immune function via circulation [46]. In this study, the significant enrichment of G-Re-downregulated genes in this pathway suggests that G-Re may inhibit its overactivation. Immunohistochemical results confirmed that G-Re significantly suppressed UVB-induced expression of COX-2, a key inflammatory enzyme, supporting its regulatory role at the protein level. The effective inhibition of this upstream key enzyme (COX-2) was accompanied by significant reductions in the levels of downstream effector cytokines, including IL-6, IL-8, and TNF-α. The decreased levels of these pro-inflammatory cytokines were also associated with attenuated pulmonary inflammation and edema, as reflected by a significantly reduced lung index in the G-Re group, as well as restored immune function in the spleen and thymus, indicated by an increased spleen index and a normalized thymus index following G-Re treatment. These results collectively suggest that ginsenoside Re may suppress the UVB-induced inflammatory cascade by targeting and inhibiting the overactivation of arachidonic acid metabolism, thereby reducing the production of pro-inflammatory mediators such as PGE2 at the source.

In summary, this study employed an integrated multi-omics approach to systematically elucidate the core mechanism by which ginsenoside Re ameliorates skin photodamage, namely, enhancing skin antioxidant capacity through regulating the glutathione metabolism pathway. Secondly, ginsenoside Re may also coordinate the repair of the skin lipid barrier and suppress inflammatory responses by modulating lipid metabolism and arachidonic acid metabolism, thereby forming a multi-target protective network. Furthermore, in in vivo rat experiments, the administration dose of ginsenoside Re used in this study (20–40 mg/kg, consecutive administration for 14 days) was compared with the reported effective doses of the established photoprotective agents lutein (570 mg/kg/day for 15 days; original literature dose: 200 μM/rat/day) [47] and anthocyanins (12.5–50 mg/kg/day for 40 days; original literature dose: 2.5–10 mg/rat/day) [48]. The results demonstrated that ginsenoside Re could exert significant photoprotective effects at a relatively lower dose. These findings provide solid theoretical support for the development of ginsenoside Re as a multi-target and high-efficacy natural skin photoprotective agent. However, this study still has certain limitations, and future research can conduct in-depth explorations targeting these aspects: First, although this study has clarified the core metabolic pathways regulated by ginsenoside Re, the precise molecular mechanism of its upstream signal transduction has not been fully elucidated. For instance, its direct action targets and activation/inhibition modes on key molecules of the Nrf2 pathway and MAPK pathway remain to be further verified [49,50,51]. Subsequent studies can adopt a combination of techniques, such as activity-based chemoproteomics to pinpoint the direct protein targets of G-Re, followed by phospho-specific immunoblotting and co-immunoprecipitation to elucidate its regulatory effects on key signaling molecules and their interactions, thereby revealing the upstream signal regulatory mechanism. Secondly, this study only verified the photoprotective effect within a specific dose range (20–40 mg/kg) and lacked systematic dose-response relationship analysis, making it impossible to determine the minimum effective dose, optimal therapeutic dose, and dose threshold of its efficacy. This limits dose optimization during clinical translation. Future research needs to conduct studies with multiple gradient dose groups and longer administration cycles to clarify the differences in efficacy at different doses and the safety of long-term application. Finally, this study was conducted based on in vivo experiments in rats, and there are differences between the animal model and the physiological environment of human skin. The safety, efficacy, and durability of its effects in human skin remain to be confirmed. Subsequent studies can construct experimental models closer to human skin, such as 3D skin tissue models, or carry out small-scale clinical pilot studies. Meanwhile, the synergistic effect of ginsenoside Re with other natural photoprotective agents can be explored to develop multi-component compound preparations for enhancing photoprotective efficacy.

## 4. Materials and Methods

### 4.1. Materials

#### 4.1.1. Chemicals and Reagents

Ginsenoside Re (G-Re, CAS: 52286-59-6, Batch No. DST241023-014, molecular weight: 947.1 g/mol, C_48_H_82_O_18_) with a purity of ≥98% as determined by HPLC was obtained from Desite Biotechnology Co., Ltd. (Chengdu, China).

#### 4.1.2. Cell Line and Animals

The human immortalized keratinocyte cell line (HaCaT) was acquired from the Central Laboratory of Xiangya School of Medicine, Central South University (Changsha, China). Male Sprague-Dawley rats (160–180 g) were supplied by Liaoning Changsheng Biotechnology Co., Ltd. (Benxi, Liaoning, China) (License No. SCXK (Liao) 2020-0001).

#### 4.1.3. Assay Kits and Antibodies

Commercial assay kits for catalase (CAT), superoxide dismutase (SOD), malondialdehyde (MDA), glutathione peroxidase (GSH-Px), and total antioxidant capacity (T-AOC) were purchased from Nanjing Jiancheng Bioengineering Institute (Nanjing, China). Enzyme-linked immunosorbent assay (ELISA) kits for interleukin-6 (IL-6), interleukin-8 (IL-8), and tumor necrosis factor-alpha (TNF-α) were obtained from Huangshi Yanke Bioengineering Co., Ltd. (Huangshi, China). The bicinchoninic acid (BCA) protein assay kit was sourced from Beyotime Biotechnology (Shanghai, China). Antibodies against CHAC1, GSTA1, GCLM, GPX1, and β-actin used for Western Blot analysis were procured from Abcam (Cambridge, UK).

#### 4.1.4. Instruments

A Philips TL 20W/101 narrow-band UVB lamp (peak wavelength: 311–313 nm) was used as the UVB radiation source. The irradiation dose was monitored and controlled in real-time using a calibrated UV-340A radiometer (Lutron, Taipei, Taiwan). A UHPLC-QE-Orbitrap-MS system (Q Exactive™ series, Thermo Fisher Scientific, Waltham, MA, USA) was employed for untargeted metabolomic detection.

### 4.2. Methods

#### 4.2.1. Cell Culture and Treatment

HaCaT cells were cultured in RPMI-1640 medium supplemented with 10% fetal bovine serum and 100 U/mL penicillin-streptomycin at 37 °C under 5% CO_2_. Cells at approximately 80% confluence during the logarithmic growth phase were utilized for experiments. G-Re was dissolved in dimethyl sulfoxide (DMSO) to prepare a 1 mg/mL stock solution, which was subsequently diluted with culture medium to working concentrations prior to use. The final DMSO concentration was maintained below 0.1% in all treatment groups, with no observed effect on cell viability. Cells were randomly allocated into three groups: normal control (NC), UVB model, and UVB + G-Re. Prior to UVB irradiation, cells were pre-incubated with G-Re at final concentrations of 0, 5, 15, 45, and 90 µg/mL (corresponding to ≈0, 5.3, 15.8, 47.5, and 95.0 μM) for 4 h. Following incubation, the medium was aspirated and cells were gently washed three times with PBS. Cells were then covered with PBS and exposed to UVB irradiation at doses of 0, 60, 120, 240, and 480 mJ/cm^2^ using a pre-warmed UVB lamp. The irradiance intensity was calibrated to 0.33 mW/cm^2^ at a vertical distance of 30 cm using a UV radiometer, with exposure duration calculated according to the formula: UVB irradiation dose (mJ/cm^2^) = UVB irradiance intensity (mW/cm^2^) × time (s). After irradiation, the PBS overlay was replaced with serum-free RPMI-1640 medium containing corresponding G-Re concentrations, and cells were cultured for an additional 24 h before subsequent analysis.

#### 4.2.2. Cell Viability Assay (MTT Method)

Cell viability was assessed using the MTT assay. Briefly, HaCaT cells were seeded in 96-well plates at a density of 5 × 10^4^ cells per well in 200 μL of complete medium and cultured for 24 h to allow complete adhesion. The cells were then subjected to drug treatment and UVB irradiation as described in Section 4.2. After an additional 24 h of incubation, 20 μL of MTT solution (5 g/L) was added to each well, followed by further incubation for 4 h. The supernatant was carefully removed, and 150 μL of dimethyl sulfoxide was added to each well to dissolve the formazan crystals by shaking at 37 °C for 10 min. Absorbance was measured at a wavelength of 570 nm using a microplate reader. Cell viability was calculated as follows: Cell viability (%) = (Absorbance of treatment group/Absorbance of blank control group) × 100%.

#### 4.2.3. Measurement of Intracellular SOD Activity and MDA Content

HaCaT cells were seeded in 6-well plates at a density of 2.5 × 10^5^ cells per well with 2 mL of complete medium and cultured until adequate adhesion was achieved. Following the procedure described in Section 4.2, cells were pretreated with G-Re at concentrations of 15, 45, and 90 µg/mL (corresponding to ≈0, 5.3, 15.8, 47.5, and 95.0 μM) and subjected to UVB irradiation. These concentrations are comparable to the effective ranges reported for other photoprotective agents in HaCaT cell models, such as caffeic acid and ferulic acid (0–60 µM) [52]. After UVB exposure, cells were cultured for an additional 24 h. Subsequently, the supernatant was aspirated, and cells were washed three times with ice-cold PBS. Then, 200 μL of RIPA lysis buffer was added to each well, and cells were lysed completely on ice. The lysate was scraped using a cell scraper, transferred to centrifuge tubes (Corning Incorporated, Corning, NY, USA), and centrifuged at 4 °C and 13,000 r/min for 30 min. The supernatant was carefully collected, aliquoted, and stored at −80 °C for subsequent analysis. Prior to measurement, samples were thawed on ice or at 2–8 °C. Total protein concentration was determined strictly in accordance with the instructions of the BCA protein quantification kit. The activities of superoxide dismutase (SOD) and the content of malondialdehyde (MDA) were measured using commercially available assay kits, following the manufacturers’ protocols.

#### 4.2.4. Animals and Drugs

Male Sprague-Dawley rats were housed in the Animal Experiment Center of Changchun University of Chinese Medicine. All housing conditions complied with the Chinese National Standard “Laboratory Animal—Requirements of Environment and Housing Facilities” (GB 14925-2001) [53], and all procedures were conducted in accordance with the guidelines approved by the Animal Ethics Committee of Changchun University of Chinese Medicine (Approval No. 2024776). The doses of Ginsenoside Re used in this study (20 mg/kg and 40 mg/kg) are consistent with the commonly reported safe dose ranges for Ginsenoside Re in rodent studies [54,55]. Ginsenoside Re was suspended in 0.5% sodium carboxymethyl cellulose solution to prepare medium- (20 mg/kg, ≈21.1 µmol/kg) and high-dose (40 mg/kg, ≈42.2 µmol/kg) formulations for intragastric administration.

#### 4.2.5. Radiation Model and Drug Administration

Rats were randomly divided into four groups (*n* = 6): control, model, low-dose G-Re, and high-dose G-Re groups. Animals in the model and G-Re-treated groups were exposed to UVB irradiation using a narrow-band UVB lamp, with the irradiation dose monitored in real-time using a calibrated radiometer. The radiation parameters were set with reference to the study by Moon et al. [56] and appropriately adjusted according to experimental conditions. The dorsal skin of rats was positioned 30–42 cm from the radiation source and subjected to a single UVB dose of 170 mJ/cm^2^ per session. which approximates one minimal erythema dose (MED). The MED is defined as the lowest ultraviolet dose required to induce a barely perceptible erythema in the skin and serves as a key metric for evaluating skin sensitivity to UV radiation and quantifying its biological effects [57]. Therefore, this dose was administered once daily for 14 consecutive days, resulting in a cumulative UVB dose of 2.38 J/cm^2^, aiming to simulate short-term repeated UV exposure and establish a significant and stable photodamage animal model. The control and model groups received daily oral administration of 0.5% sodium carboxymethyl cellulose solution, while the G-Re-treated groups were administered G-Re by gavage at corresponding doses. All treatments were conducted once daily for 14 consecutive days, with gavage administration completed 10 min prior to each irradiation session.

#### 4.2.6. Sample Collection

On day 14, rats were anesthetized via intraperitoneal injection of 10% chloral hydrate and euthanized. Dorsal skin tissues were excised and processed as follows: one portion was fixed in 10% neutral buffered formalin for histopathological examination; another portion was snap-frozen in liquid nitrogen and stored at −80 °C for subsequent transcriptome sequencing (RNA-seq). The remaining skin tissue was rinsed with ice-cold saline, minced, and homogenized. The homogenate was centrifuged at 3000 r/min for 15 min at 4 °C, and the resulting supernatant was aliquoted for the measurement of oxidative stress markers (CAT, SOD, MDA, GSH-Px, T-AOC), inflammatory cytokines (IL-6, IL-8, TNF-α), and Western Blot analysis. Major organs (heart, liver, spleen, lungs, kidneys, brain, and thymus) were harvested, weighed, and organ-to-body weight ratios were calculated. Additionally, 24-h urine samples were collected on the final day. For metabolomic analysis, proteins were precipitated by adding a five-fold volume of ice-cold acetonitrile-methanol (50:50, *v*/*v*) to the urine. After vortexing for 1 min and incubation at −20 °C for 20 min, the mixture was centrifuged at 13,000 r/min for 15 min at 4 °C. The supernatant was then collected, further centrifuged at 4000 r/min for 15 min, filtered through a 0.22 μm membrane, and subjected to LC-MS/MS analysis.

#### 4.2.7. Biochemical Analysis of Skin Tissue

Levels of antioxidant parameters, including catalase (CAT), superoxide dismutase (SOD), malondialdehyde (MDA), glutathione peroxidase (GSH-Px), and total antioxidant capacity (T-AOC), along with inflammatory cytokines (IL-6, IL-8, and TNF-α), were determined using commercial assay kits according to the manufacturers’ protocols. Protein concentrations were quantified using the bicinchoninic acid (BCA) method with a commercial protein assay kit.

#### 4.2.8. HE, Masson Staining and Immunohistochemistry

Paraffin-embedded skin sections (5 μm) were prepared and subjected to hematoxylin and eosin (H&E) staining and Masson’s trichrome staining following standard protocols to evaluate general histopathology and collagen deposition, respectively. For immunohistochemistry (IHC), sections were mounted on APES-coated slides. After microwave-assisted antigen retrieval and blocking with normal goat serum, sections were incubated overnight at 4 °C with rabbit monoclonal antibodies against COX-2 and MMP-9. Specific binding was detected using a Streptavidin-Biotin Complex (SABC) kit (ZSGB-BIO, Beijing, China), visualized with 3,3′-diaminobenzidine (DAB) chromogen, and nuclei were counterstained with hematoxylin. Negative controls were processed by replacing the primary antibody with phosphate-buffered saline (PBS). Protein expression was quantified by measuring the percentage of positive staining area (Area%) in five randomly selected fields per section using ImageJ software.

#### 4.2.9. Metabolomics Analysis

Rat urine samples from each group were separated and analyzed using UHPLC-QE-Orbitrap-MS. An Ascentis^®^ Express C18 column (3.0 mm × 50 mm, 2.7 μm) was used. Column temperature: 30 °C; the elution gradient was processed within 25 min at a flow rate of 0.3 mL/min and an injection volume of 5 μL; mobile phase A was 0.1% formic acid in water, and B was acetonitrile. The elution gradient was 95–75% A at 0–7 min; 75–30% A at 7–15 min; 30–0% A at 15–21 min; 0–95% A at 21–25 min. For mass spectrometry conditions, the Full MS-ddMS^2^ positive and negative ion switching scan mode was adopted, with a Full MS resolution of 70,000 and a dd-MS2 resolution of 35,000; scan range m/z: 100–1500, collision energy of 30 eV. The ion source was an electrospray ionization source. The ion source parameters were set as follows: sheath gas flow rate was 40 Arb, auxiliary gas flow rate was 12 Arb, ion spray voltage was set to ±3.5 KV, and capillary temperature was 350 °C. We pooled equal volumes of all samples to prepare mixed quality control (QC) samples. These QC samples were processed and determined in the same way as the analytical method to ensure their comparability. This method effectively represents the entire sample set, facilitating regular monitoring of analytical stability through intermittent injections at fixed intervals (injecting QC once every six samples). After obtaining the original mass spectrometry data, SIMCA14.1 software was used to perform principal component analysis (PCA) to evaluate the overall distribution of samples and the degree of dispersion between each group. Orthogonal partial least squares discriminant analysis (OPLS-DA) was used to screen differential metabolites, and the reliability of model fitting was tested through 200 permutation tests. The metabolites (VIP > 1.0, *p* < 0.05) were considered as potential differential metabolites. The Human Metabolome Database HMDB (https://hmdb.ca/) (accessed on 10 March 2025) was used as a tool for LC-MS identification of differential metabolites. And pathway analysis was performed using MetaboAnalyst 5.0 (https://www.metaboanalyst.ca/) (accessed on 5 April 2025).

#### 4.2.10. Transcriptomics Analysis

Four skin samples were randomly selected from each of the control group, model group, and administration group. Total RNA was extracted from the animals using TRIzol Reagent (Life Technologies, Carlsbad, CA, USA) according to the manufacturer’s instructions. RNA concentration and purity were measured with a NanoDrop 2000 (Thermo Fisher Scientific, Wilmington, DE, USA). RNA integrity was assessed using the RNA Nano 6000 Assay Kit on an Agilent Bioanalyzer 2100 system (Agilent Technologies, Santa Clara, CA, USA). Sequencing libraries were generated using the Hieff NGS Ultima Dual-mode mRNA Library Prep Kit for Illumina (Yeasen Biotechnology (Shanghai) Co., Ltd., Shanghai, China), and library quality was evaluated using the Agilent Bioanalyzer 2100 system. Libraries were sequenced on the Illumina NovaSeq platform. Differential genes were further screened and analyzed, followed by enrichment and clustering analyses of the samples. The thresholds for significantly differential expression were set as *p* < 0.05, VIP > 1, and |log_2_FC| > 1. GO and KEGG pathway enrichment analyses of DEGs were both performed using the DAVID database (https://davidbioinformatics.nih.gov/) (accessed on 2 June 2025).

#### 4.2.11. Integration of Transcriptomics and Metabolomics

The joint pathway analysis was performed using the MetaboAnalyst 5.0 platform. Significantly altered metabolites (VIP > 1.0, *p* < 0.05) and genes (|log_2_FC| > 1, *p*-adjusted < 0.05) were uploaded for over-representation analysis (ORA) based on the KEGG pathway database. Pathways with a hypergeometric test *p*-value < 0.05 were considered significantly enriched.

#### 4.2.12. Western Blot Analysis

Total proteins were extracted from skin tissues by homogenization in RIPA lysis buffer containing a protease inhibitor cocktail. The homogenates were centrifuged at 12,000× *g* for 15 min at 4 °C to collect the supernatant. Protein concentrations were determined using a bicinchoninic acid (BCA) protein assay kit. Equal amounts of protein (30 μg per lane) were separated by SDS-PAGE and subsequently transferred onto PVDF membranes. After blocking with 5% non-fat milk in TBST for 1 h at room temperature, the membranes were incubated overnight at 4 °C with specific primary antibodies diluted at 1:1000. Following primary antibody incubation, the membranes were incubated with horseradish peroxidase (HRP)-conjugated secondary antibodies (1:5000) for 1.5 h at room temperature. Protein bands were detected using an enhanced chemiluminescence (ECL) substrate and imaged with a ChemiDOC MP Imaging System. The band intensity was quantified by densitometric analysis using ImageJ software (version 1.54), and the relative protein expression levels were normalized to β-actin.

#### 4.2.13. Molecular Docking

To investigate the potential interactions between ginsenoside Re and key proteins in the glutathione metabolic pathway (ChAC1, GSTA1, GCLM, GPX1) at the structural level, molecular docking studies were performed. The three-dimensional structure of ginsenoside Re was obtained from the TCMSP database (https://www.tcmsp-e.com/) (accessed on 5 August 2025). The crystal structures of the target proteins were retrieved from the UniProt database and the AlphaFold Protein Structure Database (https://alphafold.ebi.ac.uk/) (accessed on 5 August 2025). Protein preparation included dehydration using PyMOL (version 2.5.4), followed by the addition of charges and hydrogen atoms with AutoDock (version 1.2.3) tools. Molecular docking was carried out using AutoDock Vina to determine binding energies, and the complex with the strongest binding affinity was selected for further analysis. Visualization of the docking modes was performed using PyMOL.

#### 4.2.14. Statistical Analysis

Data were imported into GraphPad Prism 7.0 for further analysis and graph generation. Results are expressed as the mean ± standard deviation (SD). Statistical significance among multiple groups was determined using one-way analysis of variance (ANOVA) followed by Tukey’s post-hoc test for pairwise comparisons. A *p*-value < 0.05 was considered statistically significant.

## 5. Conclusions

Based on integrated histopathological, biochemical, transcriptomic, and metabolomic analyses, this study demonstrates that ginsenoside Re significantly ameliorates UVB-induced skin photodamage. The protective effects are primarily mediated through the enhancement of cutaneous antioxidant capacity via regulation of the glutathione metabolism pathway. Additionally, ginsenoside Re may coordinately promote skin barrier restoration and attenuate inflammatory responses by modulating multiple pathways, including lipid metabolism, arachidonic acid metabolism, and histidine metabolism. These findings highlight ginsenoside Re as a promising natural photoprotective agent characterized by favorable biocompatibility, skin permeability, and multi-pathway synergistic activity, thereby providing a theoretical foundation for developing novel strategies to prevent and repair UV-induced skin damage.

## Figures and Tables

**Figure 1 ijms-27-00708-f001:**
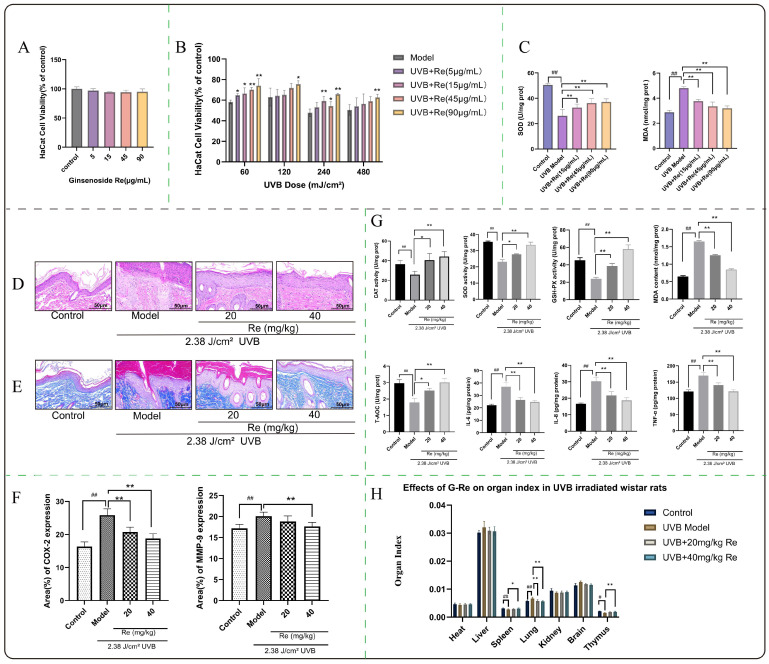
Protective effect of ginsenoside Re (G-Re) against UVB-induced skin photodamage in vivo and vitro. (**A**) Viability effects of G-Re on Normal HaCaT cells. (**B**) Viability effects of G-Re with various concentrations on HaCaT cells cultured for 24 h after different dosages of UVB irradiation. (**C**) Effects of G-Re on SOD activity and MDA content in HaCaT cells after UVB irradiation. (**D**) H&E staining of rat skin tissues. (Magnification: ×200) (**E**) Masson Staining of rat skin tissues. (Magnification: ×200) (**F**) Immunohistochemistry (IHC) quantification of COX-2 and MMP-9 positive expression in rat skin tissues, presented as the percentage of positive area (Area%). (**G**) Changes in CAT, SOD, GSH-Px, MDA, T-AOC, IL-6, IL-8, and TNF-α levels in rat skin. (**H**) Assessment of systemic effects as indicated by organ-to-body weight ratios. The data are presented as mean ± SD (*n* = 6). # *p* < 0.05, ## *p* < 0.01 compared with control group. * *p* < 0.05, ** *p* < 0.01 compared with model group. The molecular weight of G-Re is 947.1 g/mol, which could be converted into unit molar mass and presented as follows: 1 μg G-Re ≈ 1.06 nmol, and 1 mg G-Re ≈ 1.06 μmol.

**Figure 2 ijms-27-00708-f002:**
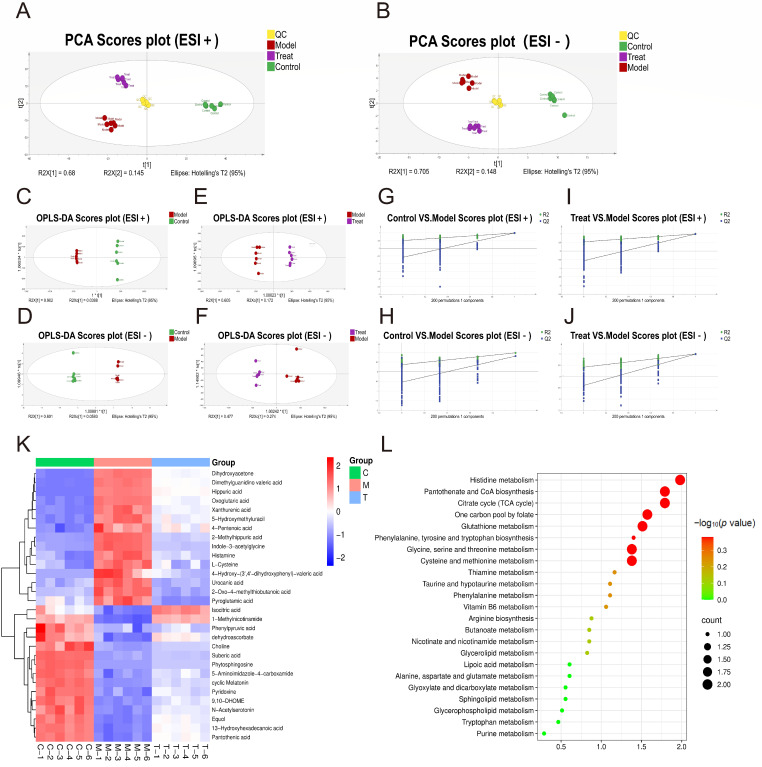
Urinary metabolomics analysis (LC-MS/MS) of UVB-induced photodamaged rats after ginsenoside Re intervention. (**A**,**B**) PCA score plots in positive and negative ion modes. (**C**–**F**) OPLS-DA score plots in positive and negative ion modes. (**G**–**J**): OPLS-DA permutation test plots in positive and negative ion modes. (**K**) Hierarchical clustering heatmap of differential metabolites. (**L**) KEGG enrichment analysis plot of differential metabolomics.

**Figure 3 ijms-27-00708-f003:**
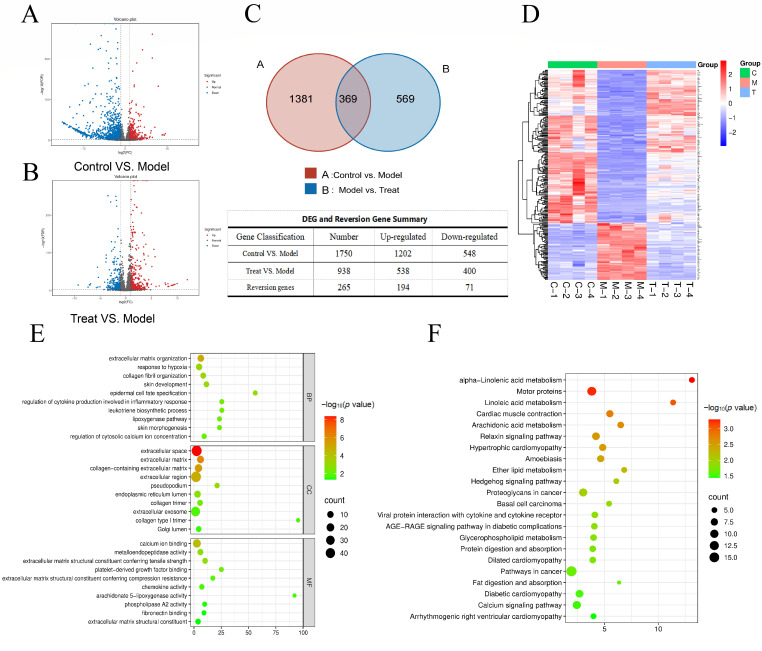
Skin transcriptomic analysis of UVB-induced photodamaged rats after ginsenoside Re intervention. (**A**) Volcano plot of differentially expressed genes (DEGs) between the Control and Model groups. (**B**) Volcano plot of DEGs between the Ginsenoside Re-treated (40 mg/kg) and Model groups. For A & B: Red/blue dots represent significantly up-/down-regulated genes (thresholds: |log_2_FC| > 1, *p* < 0.05). (**C**) Venn diagram of DEG overlap and summary. The diagram shows 369 common DEGs, with 265 reversed by ginsenoside Re. (**D**) Hierarchical clustering heatmap of the 265 reverted DEGs. Columns: C (Control Group), M (UVB Model Group), T (Ginsenoside Re-treated Group, 40 mg/kg); colors indicate expression levels relative to the mean (red: high, blue: low). (**E**) Gene Ontology (GO) enrichment analysis. Enriched terms are categorized into the three main GO domains: Biological Process (BP), Cellular Component (CC), and Molecular Function (MF). (**F**) KEGG pathway enrichment analysis; bubble size indicates gene count, color indicates enrichment significance (−log_10_(*p*-value)).

**Figure 4 ijms-27-00708-f004:**
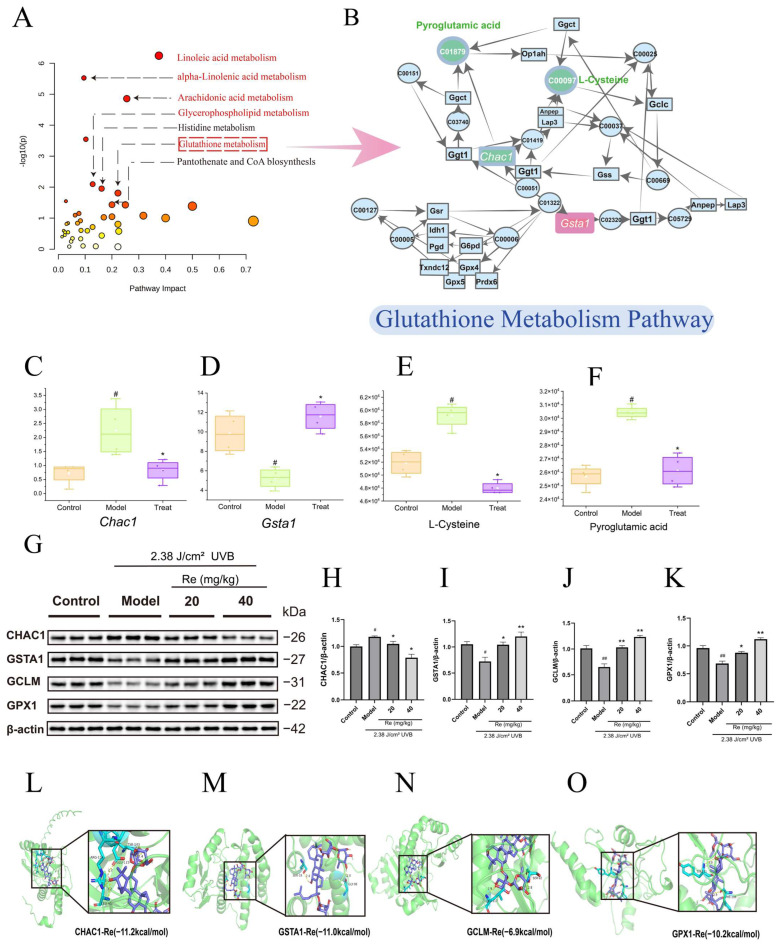
Integrated multi-omics and experimental validation of the glutathione metabolism pathway in ginsenoside Re-mediated protection against UVB-induced skin photodamage. (**A**) Integrated transcriptome-metabolome KEGG enrichment plot. (**B**) Schematic diagram of glutathione metabolism pathway regulation. (Red: Up-regulated; Green: Down-regulated) (**C**–**F**) The quantitative analysis for the DEGs (*Chac1*, *Gsta1*) and metabolites (L-Cysteine, Pyroglutamic acid). (**G**) Electrophoretic diagram of different groups. (**H**–**K**) Expressions of protein level of CHAC1, GSTA1, GCLM and GPX1. (**L**–**O**) Molecular docking of CHAC1-Re, GSTA1-Re, GCLM-Re and GPX1-Re. (*n* = 3) # *p* < 0.05, ## *p* < 0.01 compared with control group. * *p* < 0.05, ** *p* < 0.01 compared with model group.

**Table 1 ijms-27-00708-t001:** Differential Metabolite Information in Urine.

Metabolite	Formula	HMDB ID	M/Z	RT(Min)	Mode	PPM	Change Trend	
							C/M	T/M
Dehydroascorbate	C_6_H_8_O_7_	HMDB0304326	215.0177	1.57	Pos	7	Up	Up
1-Methylnicotinamide	C_7_H_9_N_2_O	HMDB0000699	137.0721	1.08	Pos	4	Up	Up
13-Hydroxyhexadecanoic acid	C_16_H_32_O_3_	HMDB0112191	295.2284	16.05	Pos	2	Up	Up
Phytosphingosine	C_18_H_39_NO_3_	HMDB0004610	318.2993	16.36	Pos	3	Up	Up
Choline	C_5_H_14_NO	HMDB0000097	104.1082	1.08	Pos	6	Up	Up
Isocitric acid	C_6_H_8_O_7_	HMDB0000193	407.0458	1.57	Pos	6	Up	Up
Phenylpyruvic acid	C_9_H_8_O_3_	HMDB0000205	165.0521	27.96	Pos	1	Up	Up
5-Aminoimidazole-4-carboxamide	C_4_H_6_N_4_O	HMDB0003192	253.1175	1.39	Pos	8	Up	Up
Equol	C_15_H_14_O_3_	HMDB0002209	243.1048	1.45	Pos	13	Up	Up
Pantothenic acid	C_9_H_17_NO_5_	HMDB0000210	242.1014	1.45	Pos	6	Up	Up
cyclic Melatonin	C_13_H_14_N_2_O_2_	HMDB0060811	461.2135	1.45	Pos	10	Up	Up
Pyridoxine	C_8_H_11_NO_3_	HMDB0000239	170.0824	1.22	Pos	7	Up	Up
N-Acetylserotonin	C_12_H_14_N_2_O_2_	HMDB0001238	219.1143	1.48	Pos	7	Up	Up
Suberic acid	C_8_H_14_O_4_	HMDB0000893	349.1859	13.56	Pos	1	Up	Up
9,10-DHOME	C_18_H_34_O_4_	HMDB0010221	313.2411	1.3	Neg	2	Up	Up
4-Pentenoic acid	C_5_H_8_O_2_	HMDB0031602	123.0413	0.9	Pos	3	Down	Down
4-Hydroxy-(3′,4′-dihydroxyphenyl)-valeric acid	C_11_H_14_O_5_	HMDB0041679	219.1008	25.36	Pos	11	Down	Down
Dimethylguanidino valeric acid	C_8_H_15_N_3_O_3_	HMDB0240212	202.1199	1.14	Pos	6	Down	Down
2-Methylhippuric acid	C_10_H_11_NO_3_	HMDB0011723	216.0647	15.27	Pos	7	Down	Down
Pyroglutamic acid	C_5_H_7_NO_3_	HMDB0000267	259.0941	1.47	Pos	6	Down	Down
Dihydroxyacetone	C_3_H_6_O_3_	HMDB0001882	203.052	12.64	Pos	3	Down	Down
Hippuric acid	C_9_H_9_NO_3_	HMDB0000714	180.0666	12.64	Pos	6	Down	Down
Xanthurenic acid	C_10_H_7_NO_4_	HMDB0000881	206.0463	1.61	Pos	7	Down	Down
Indole-3-acetylglycine	C_12_H_12_N_2_O_3_	HMDB0240661	233.0936	18.32	Pos	7	Down	Down
Oxoglutaric acid	C_5_H_6_O_5_	HMDB0000208	293.0487	0.93	Pos	6	Down	Down
Urocanic acid	C_6_H_6_N_2_O_2_	HMDB0000301	139.0512	1.26	Pos	7	Down	Down
2-Oxo-4-methylthiobutanoic acid	C_5_H_8_O_3_S	HMDB0001553	166.0508	1.35	Pos	15	Down	Down
5-Hydroxymethyluracil	C_5_H_6_N_2_O_3_	HMDB0000469	307.0645	0.93	Pos	1	Down	Down
Histamine	C_5_H_9_N_3_	HMDB0000870	245.1509	1.52	Pos	10	Down	Down
L-Cysteine	C_3_H_7_NO_2_S	HMDB0000574	243.0459	20.74	Pos	4	Down	Down

Note: C/M = Control vs. Model; T/M = Treat (40 mg/kg Ginsenoside Re) vs. Model. ‘Up’ indicates metabolite level higher than the Model group; ‘Down’ indicates lower than the Model group.

## Data Availability

The raw RNA-seq data generated in this study have been deposited in the NCBI Sequence Read Archive (SRA) under BioProject accession number PRJNA1346042. The raw metabolomics data have been deposited to the MetaboLights database under the accession number MTBLS13149.

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
