# Peer review of "Ginsenoside Re Ameliorates UVB-Induced Skin Photodamage by Modulating the Glutathione Metabolism Pathway: Insights from Integrated Transcriptomic and Metabolomic Analyses"

_ijms, 2026, doi:10.3390/ijms27020708_

Round 1
Reviewer 1 Report
Comments and Suggestions for Authors
The manuscript “Ginsenoside Re Ameliorates UVB-Induced Skin Photodamage by Modulating the Glutathione Metabolism Pathway: Insights from Integrated Transcriptomic and Metabolomic Analyses” presents relevant and well-executed experiments that provide insights into the photoprotective role of ginsenoside Re. However, several points should be addressed to improve clarity, coherence, and scientific rigor:
- In the abstract, include the generic name of ginseng and italicize the scientific name.
- In the abstract, avoid using “etc.”—instead, name the most relevant analyses performed.
- Since “ultraviolet” is abbreviated as UV upon first mention, ensure consistent use of “UV” throughout the manuscript.
- There are multiple typographical errors such as inconsistent spacing between words and punctuation marks. A detailed proofreading is necessary.
- In Figures 1A and 1B, the y-axis label is marked as “%”, which appears unclear. If control viability is close to 100%, the graph should reflect that. Please verify whether the current label is accurate.
- I recommend converting the concentrations of ginsenoside Re to molarity units (e.g., μM or mM). Since it is a pure compound, molarity provides a more standardized and interpretable measure.
- Provide biological context for the experimental doses. What concentrations of UV filters are typically achieved in the skin when commercial products are used? What are the average UVB exposure levels under critical environmental conditions? This would help readers assess the physiological relevance of the experimental design.
- Before presenting the results shown in Figure 1C, briefly introduce the purpose of that experiment.
- Define SOD (superoxide dismutase) and MDA (malondialdehyde) at their first mention in the text to improve readability.
- The molecular docking section appears disconnected from the rest of the manuscript. While the study shows that ginsenoside Re influences the expression of redox-related enzymes, the docking analysis assumes a direct interaction that is not supported by the presented data.
- The manuscript does not clarify whether the predicted binding would imply enzyme activation, inhibition, or no functional effect. Since protein expression changes could be indirect or downstream of other pathways, the docking results appear speculative.
- Unless a clear rationale is provided for the docking targets and the expected binding effects, I recommend removing the molecular docking results from the manuscript.
- I suggest including a paragraph discussing the limitations of the current study and outlining future directions, such as deeper dose–response analyses, or mechanistic dissection of upstream signaling pathways.
Author Response
Dear reviewer.
Thank you very much for taking the time to review this manuscript. Please find the detailed responses below and the corresponding revisions in track changes in the re-submitted files.

Reviewer 2 Report
Comments and Suggestions for Authors
This manuscript evaluated the effect of ginsenoside Re against UVB-induced skin photodamage.
Integreated transcriptomic and metabolomic analyses with multi-omics integration and Western Blotting yielded positive results.
I would like to recommend this manuscript accept.
Minor corrections are:
- Treatment concentration better be expressed in ug/mL instead of mg/L.
- In Fig.1 C, dose-dependent effect is not significant. Please expain. In addition, statistical results should be marked between samples.
Author Response

(The authors gave the same response as above.)
